# A Familial Novel Putative-Pathogenic Mutation Identified in Plaque-Psoriasis by a Multigene Panel Analysis

**DOI:** 10.3390/ijms24054743

**Published:** 2023-03-01

**Authors:** Marcella Nunziato, Anna Balato, Anna Ruocco, Valeria D’Argenio, Roberta Di Caprio, Nicola Balato, Fabio Ayala, Francesco Salvatore

**Affiliations:** 1CEINGE-Biotecnologie Avanzate Franco Salvatore, Via Gaetano Salvatore, 486, 80145 Naples, Italy; 2Department of Molecular Medicine and Medical Biotechnologies, University of Naples Federico II, Via Sergio Pansini, 5, 80131 Naples, Italy; 3Dermatology Unit, University of Campania “Luigi Vanvitelli”, 80131 Naples, Italy; 4Department of Human Sciences and Quality of Life Promotion, San Raffaele Open University, 00166 Roma, Italy; 5Microbiology and Virology Unit, Cotugno Hospital, AORN Dei Colli, Via Gaetano Quagliariello, 54, 80131 Naples, Italy; 6Italian “School of Psoriasis” Association, 81031 Aversa, Italy; 7Department of Clinical Medicine and Surgery, University of Naples Federico II, Via Sergio Pansini, 5, 80131 Naples, Italy

**Keywords:** multigene panel, plaque-psoriasis, predictive medicine, psoriasis familial mutations, predisposing genes

## Abstract

Psoriasis is a chronic multifactorial skin disorder with an immune basis. It is characterized by patches of skin that are usually red, flaky and crusty, and that often release silvery scales. The patches appear predominantly on the elbows, knees, scalp and lower back, although they may also appear on other body areas and severity may be variable. The majority of patients (about 90%) present small patches known as “plaque psoriasis”. The roles of environmental triggers such as stress, mechanical trauma and streptococcal infections are well described in psoriasis onset, but much effort is still needed to unravel the genetic component. The principal aim of this study was to use a next-generation sequencing technologies-based approach together with a 96 customized multigene panel in the attempt to determine if there are germline alterations that can explain the onset of the disease, and thus to find associations between genotypes and phenotypes. To this aim, we analyzed a family in which the mother showed mild psoriasis, and her 31-year-old daughter had suffered from psoriasis for several years, whereas an unaffected sister served as a negative control. We found variants already associated directly to psoriasis in the *TRAF3IP2* gene, and interestingly we found a missense variant in the *NAT9* gene. The use of multigene panels in such a complex pathology such as psoriasis can be of great help in identifying new susceptibility genes, and in being able to make early diagnoses especially in families with affected subjects.

## 1. Introduction

Psoriasis is a chronic immune-mediated inflammatory disease that presents with characteristic cutaneous alterations and systemic manifestations. Psoriasis has been primarily defined as an autoimmune, T-cell-mediated disease with a dysregulated inflammatory response that is composed of both innate and adaptive immunity [1,2]. Although it is preferentially a skin disease, recent studies highlight the association with several different multiorgan comorbidities implicated in the systemic inflammation [3,4,5,6]. Advances in molecular and cellular research have enabled the immunological features of psoriasis to be deciphered and helped to classify this chronic inflammatory skin disorder as the prototypic IL-23 and IL-17-dominated disease in humans [7]. The prevalence of psoriasis varies considerably among populations. Caucasians are more frequently affected than are the African-American and Oriental populations, while Mediterranean populations are the least affected. These differences may depend on ethnicity, and genetic and environmental factors [5,8,9]. Psoriasis onset is between 15 and 35 years of age and psoriasis type I can be distinguished from type II. Psoriasis type I has an early onset (<40 years), is often associated with a familial disease history and is highly associated with the human leukocyte antigen HLA-C*06:02 allele; whereas, type II develops after the age of 40 and both familiarity and severity are less frequent [10].

Disease classification is based on clinical appearance, which mainly differs depending on localization and morphology. There are four main forms of psoriasis: plaque-type, guttate, generalized pustular psoriasis and erythroderma [11,12]. Plaque-type psoriasis, which occurs in 85–90% of affected patients, is the most common form of psoriasis and it is characterized by oval or irregularly shaped, red, sharply demarcated raised plaques covered by silvery scales. The latter are accompanied by itching that usually affects the elbows, knees, scalp and lower back, but can appear anywhere on the body surface with different degrees of severity as measured by the Psoriasis Area and Severity Index (PASI), Body Surface Area, and/or the Physician’s Global Assessment (PGA) [13]. It is now well established that psoriasis is a multi-factorial disease, the onset of which is characterized by “triggering factors”. The most well-known triggering factors are physical traumas (Koebner phenomenon), infectious causes (Streptococcus), pharmacological factors (lithium, β-blockers, non-steroidal anti-inflammatory drugs—NSAIDs), endocrine-metabolic factors and lifestyle [14]. Conversely, genetic causes are not well studied and are currently of considerable interest [15,16]. Thanks to the advent of Next Generation Sequencing (NGS)-based approaches such as metagenomics, transcriptomics, and the sequencing of complete genomes, etc. [17,18], plus the use of multigene panels, also for a very complex disease such as psoriasis, many genes can render an individual more susceptible to the development of the disease in association with environmental/lifestyle factors [15,16,17,18,19]. Several gene loci are associated with psoriasis, such as HLA-Cw6 and PSORS1-9, providing initial evidence of a possibly (auto) immune component [20]. To date, 80 loci have already been identified, 47 of which through genome-wide studies (GWAS), 15 with ImmunoChip and 1 through whole exome sequencing (WES) [2,19,21,22,23]. Candidate genes are implicated, in particular, in immune mechanisms, immune system regulation pathways, in immune synapse formation, in solute transport and in inflammatory processes. Other genes that can be considered associated to psoriasis are essential for keratinocyte structural integrity and for epidermal cell homeostasis, and some are involved in epidermal differentiation [24,25,26]. The biological consequences of the genetic variations in most genes associated with psoriasis are uncertain. Given that respective psoriasis-associated single nucleotide polymorphisms (SNPs) are mainly located in non-coding regions, including introns and intergenic regions; the elucidation of their functional consequences has proven to be very challenging.

Herein, we used a 96-multigene panel related to psoriasis risk onset to analyze a family constituted by a mother affected by psoriasis and two daughters, one of whom was also affected by psoriasis while the other healthy daughter served as a family control. The analysis highlighted interesting variants that may be implicated in psoriasis onset, including one in the NAT9 gene. The latter variant is not reported in the ClinVar database nor is it associated with psoriasis in the GWAS catalog, and lastly, to our knowledge, it does not appear in any other predisposition studies. Consequently, it appears to be a novel variant related to psoriasis. The aim of this study was to identify new variants that may play a role in the onset of this complex disease.

## 2. Results and Discussion

### 2.1. NGS Data Analysis

The enrichment of each sample sequence was carried out using the Illumina BaseSpace platform. Each sample was aligned to a custom manifest file. Total aligned bases reached 100 million in the sequence regions indicated above. The mean region coverage depth in each patient exceeded 220× and the percentage of aligned reads was between 99.3% and 99.4%. The total number of SNVs in the dataset that passed quality filters was more than 3000, whereas there were 456 variants in each coding region (Table 1).

### 2.2. Identification of Variants in the Family

The rs33980500, c.28G>A (p.D10N) variant in the *TRAF3IP2* gene was found in the mother and in her daughter, both of whom were affected by mild psoriasis with a PASI Index between 3 and 4. The variant was in a heterozygous state, and it was found also in the unaffected family control. The *TRAF3IP2* gene (OMIM: 607043) is located on chromosome 6q21 and is involved in the mechanisms of the immune response of IL-17-mediated T cells which appear to play a key role in the chronic inflammatory processes that develop in subjects with psoriasis and psoriatic arthritis [27].

The missense variant rs33980500 leads to an amino acid substitution of an aspartic acid residue with asparagine (p.D10N), thereby resulting in a change of a negatively charged amino acid into a non-polar state [28]. This substitution may influence the structure of the protein and therefore also its functional activity [16]. 

The other interesting variant we found is rs751674419, c.128C>T (p.S43L) in the *NAT9* gene which is not reported in the ClinVar database. We found this variant in heterozygosity in our two affected subjects. The *NAT9* gene is located on chromosome 17q25 in the psoriasis susceptibility locus PSORS2, and encodes N-acetyltransferase 9 that is involved in the regulation of signal transduction and T cell growth [29]. The variant is a missense alteration (p.S43L) with a high Combined Annotation Dependent Depletion score (CADD score) of 29 and many pathogenic predictions, namely BayesDel, Mutation assessor and FATHMM-MKL. Before this paper, the ACMG in silico-based classification indicates it as a variant of unknown significance.

In the family where the variant was found, both the mother and daughter were affected by plaque psoriasis. Both showed the first signs of the disease in the first two decades of life: the mother at the age of 20 years and the daughter at the age of 18 years. The PASI score associated with their phenotype was mild: PASI 3 and 4, respectively. DNA sequencing showed that both mother and daughter carry the missense variant rs33980500 in the *TRAF3IP2* gene in heterozygosity, and the variant was already known to predispose for psoriasis. The same variant was searched for by direct Sanger sequencing also in the daughter not affected by psoriasis, and also, she was found to be a carrier of the variant in heterozygosity. In addition to the variant in the *TRAF3IP2* gene, both the affected mother and her affected daughter were found to be carriers of the rs751674419 variant in the *NAT9* gene.

The presence of the variant in the *NAT9* gene was checked in the family control subject in order to determine whether there was a segregation of the variant only in the affected members or, as in the case of *TRAF3IP2*, also in unaffected ones. The Sanger sequencing revealed that the daughter without psoriasis was wild type for the *NAT9* gene and, therefore, does not share the rs751674419 variant with her sister or her mother who were both affected by the disease (Figure 1).

The data obtained are of considerable interest because they suggest that the variant in the *NAT9* gene may be an additional predisposing factor, without which the mere presence of the variant in *TRAF3IP2* alone would not be sufficient for the development of the disease [30] in the studied family.

To date, the rs751674419 variant has never been found to be related to psoriasis. However, given the location of the gene in a locus already associated with psoriasis and the fact that the latter plays a role in the growth mechanisms of immunity cells, such as T cells, it could be assumed that alterations in this gene may play a role in the pathogenesis of psoriasis at an immune level [31]. The missense variant in the NAT9 gene has been found only in subjects affected by psoriasis but not in the healthy control subject of the same family.

## 3. Materials and Methods

### 3.1. Patient Selection

We selected a family composed of a 52-year-old mother affected by mild plaque psoriasis since she was 20 years old and her 31-year-old daughter who was also affected by mild plaque psoriasis since she was 18 years old. Furthermore, the sister not affected by psoriasis at the time of the analysis served as a control of the variants found. The family was enrolled by the Psoriasis Care Center of the Outpatient Clinic of the Section of Dermatology, University of Naples Federico II, Naples, Italy. The severity of psoriasis was assessed by physicians using the PASI score that is a validated and widely used tool for measuring psoriasis severity. The scale evaluates four areas of the body (head/neck, upper limbs, trunk, and lower limbs) for erythema, scaliness and thickness of psoriatic plaques. The PASI score can range from 0 to 72, with higher scores indicating greater severity. To prevent rate biases, the dermatologists who evaluated the PASI score were blinded to the design of the study. The affected proband had a PASI score of 3, and her daughter had a PASI score of 4 (Figure 2). All patients gave their written informed consent to the study prior to blood sampling, which was carried out according to the tenets of the Helsinki Declaration and approved by the University of Naples Federico II Ethics Committee (protocol number 10/15).

### 3.2. Custom Panel Design

To identify possible associations and new predisposing variants, and thanks to our experience in NGS-based approaches using multigene panels [32,33,34], we designed a panel of 96 genes that can be assembled in 6 groups: 46 genes are implicated in the immune response and inflammatory processes [35], 11 genes exert antimicrobial antiviral activity [36], 11 play a role in skin corneous layer differentiation [37], 9 are implicated in cellular adhesion and growth, 11 genes play a role in metabolism and signaling [3] and 8 are psoriasis-related genes reported to be associated with the disease (Appendix A) [27,29]. For each gene, we included the coding regions, 100 bp in each intronic boundary, the promoter and the 3’ UTR regions for a total target size of about 1 Mb and a predicted target gene coverage of 96.5%.

### 3.3. Sample Preparation and Sequencing

Genomic DNA was obtained from peripheral blood samples using the Nucleon BACC3 Genomic DNA Extraction Kit (GE Healthcare, Little Chalfont, UK), according to the manufacturer’s instructions. The first step was the sonication of human DNA. Thus, samples were dissolved in a sonication buffer of TE (10 mM Tris, 1 mM EDTA), pH 8 with a DNA concentration of 11 ng/μL and a final volume of 110 μL. The 110 μL of genomic DNA was loaded into the microtube AFA Snap-Cap and shared to a target peak size of 150–200 bp using the Covaris S220 focused-ultrasonicator (Covaris, Woburn, MA, USA) according to the manufacturer’s instructions. DNA fragment size and quantity were evaluated using the Agilent Bioanalyzer High Sensitivity DNA chip (Agilent Technologies, Santa Clara, CA, USA) and the Quant-iT™ PicoGreen^®^ dsDNA Reagent and Kits (Molecular probes, Cat. No: P11496) on the Lightcycler 480 II system (Roche), respectively. Sample library preparation and whole exome solution were captured. The psoriasis enrichment panel design was generated using the online NimbleDesign tool available at Roche website. Library preparation was performed with the KAPA Library Preparation Kit for Illumina platforms (Kapa Biosystems, KR0935—v1.14, Woburn, MA, USA) and Roche NimbleGen SeqCap EZ Library SR kit (Roche, Basel, Switzerland). Then, 1 µg of input fragmented dsDNA was used, and the KAPA standard protocol was followed. The 24 DNA samples were divided into 2 groups of 12 each for Adaptor Ligation A and B. The adapter-ligated sample was purified using Agencourt AMPure XP beads (Beckman Coulter, Brea, CA, USA) and analyzed on a Bioanalyzer DNA 1000 chip. Twenty microliters of the sample library were used for pre-capture PCR according to the NimbleGen SeqCap protocol. The PCR products were purified using Agencourt AMPure XP beads and analyzed on a Bioanalyzer DNA1000 chip. Then, the PCR products were pooled together equimolarly, and one microgram of the sample total was prepared for the hybridization with the capture baits, and the sample was hybridized for 72 h at 47 °C, and captured with the Capture Beads Plus Bound DNA. After hybridization and capture of the DNA with capture beads, the captured yield was measured using a quantitative PCR. Sequencing was carried out using the MiSeq sequencing platform (Illumina, Inc., San Diego, CA, USA). Libraries were sequenced with PE v2 (250 × 2) MiSeq Reagent Kits (For the entire workflow see Appendix A). Therefore, 8 pM of denatured libraries were combined to 5% of 8 pM PhiX and loaded into the MiSeq v2 reagent cartridge. 

### 3.4. Bioinformatic Analysis

The bioinformatic analysis was carried out using the Illumina VariantStudio data analysis software (Illumina, Inc., San Diego, CA, USA). The obtained fastq files and a custom targeted regions manifest file were uploaded on BaseSpace software (https://basespace.illumina.com/dashboard, accessed on 1 January 2022). Samples were enriched using an Enrichment Application (BWA Enrichment by Illumina, Inc. v.2.1.0.0). The workflow consists of the following major steps: reads were aligned against the custom manifest file; then, the annotation of variants was carried out using Illumina Annotation Engine. The resultant VCF file for each sample was uploaded into VariantStudio. Causative or possibly interesting variants were confirmed by Sanger sequencing.

## 4. Conclusions and Limitations

The results reported herein show that the use of NGS sequencing platforms can shed light on the weight that an individual’s genetics has in developing multigenic and multifactorial pathologies such as psoriasis, which is characterized by extreme inter- and intra-individual genetic variability. The use of multigene panels enables one to take a broad look at the genetic background of an individual by providing information about personalized therapies, or to implement actions aimed at preventing the development of comorbidities associated with diseases such as cardiovascular diseases.

A limitation of this study is that genetic data alone cannot provide all the answers to the questions on the etiopathogenesis of the disease. In fact, these data should be integrated with information relating to the environmental and social context in which the individual lives, and to co-morbidities known to be strictly associated with psoriasis.

In this context, the present study helps to shed light on the possibility that genetic alterations may be the cause of the onset of psoriasis, or may cooperate with the pathogenesis of the disease itself, together with other multifactorial factors that characterize this complex disease.

## Figures and Tables

**Figure 1 ijms-24-04743-f001:**
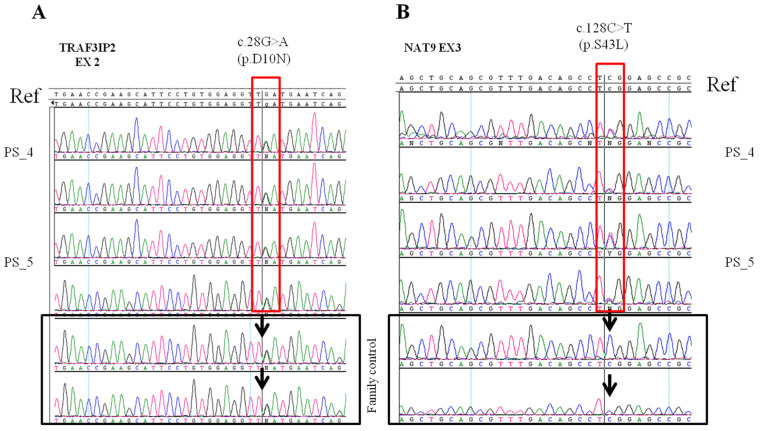
The electropherograms obtained by Sanger sequencing; for each patient, the forward (FW) and the reverse (RW) strands of the same sequence are represented. (**A**) The electropherograms of the variant in the *TRAF3IP2* gene in PS_4, PS_5 and in the familial control. All three are carriers of the variant in heterozygosity. (**B**) The electropherograms of PS_4 and PS_5 and from the family control are shown. In this case, the variant in *NAT9* is present in heterozygosity in both the mother (PS_5) and daughter (PS_4) affected by plaque-psoriasis, but not in the unaffected control (II.1) at the time of the test. Black arrows identify the presence/absence of the variant in the family negative control.

**Figure 2 ijms-24-04743-f002:**
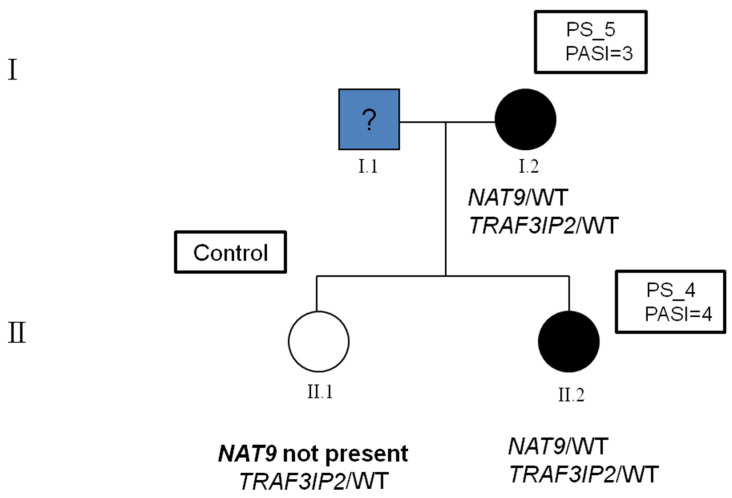
The pedigree of the family under examination and the variants found. The mother (PS_5 and PASI = 3) (proband) contracted plaque-psoriasis at 20 years of age and was found to carry both variants in *TRAF3IP2* and *NAT9* in a heterozygous status. Her daughter (PS_4 and PASI = 4) was affected by plaque-psoriasis at the age of 18 years and she carried the same two variants as her mother. The presence of the two variants was also checked in the family control (the non-affected daughter) and she was found to carry in the variant in *TRAF3IP2* in heterozygous status but was wild type for the variant in the *NAT9* gene.

**Table 1 ijms-24-04743-t001:** Raw sequencing data of the mother and her daughter.

ID NGS	Mean Region Coverage Depth	Target Coverage at 50×	Percent Aligned Reads	Variants in Genes Presented in the Panel	Variants in Coding Regions
PS_4	271.9	97.50%	99.30%	3135	456
PS_5	222	96.60%	99.40%	3048	456

## Data Availability

Our data are available by a request at salvator@unina.it.

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
