# Peer review of "A Familial Novel Putative-Pathogenic Mutation Identified in Plaque-Psoriasis by a Multigene Panel Analysis"

_ijms, 2023, doi:10.3390/ijms24054743_

Round 1

Reviewer 1 Report

Dear Authors

I read your manuscript with great interest.

I find the manuscript interesting. The introduction part needs to be improved, especially the first paragrpah in respect to english language. Otherwise no comments.

Author Response

Point-by-Point

A familial novel putative-pathogenically mutation identified in plaque-psoriasis by a multigene panel analysis

Marcella Nunziato, Anna Balato, Anna Ruocco, Valeria D'Argenio, Roberta Di Caprio, Nicola Balato, Fabio Ayala, Francesco Salvatore

Reviewer #1

Dear Authors,

I read your manuscript with great interest.

I find the manuscript interesting. The introduction part needs to be improved, especially the first

paragrpah in respect to english language. Otherwise no comments.

Reply: Thank you very much for your comments that have enabled us to improve our work. We have largely rewritten the Introduction with more informative sentences and changing some literature references. Furthermore, we have also added a final sentence at the end of the introduction. All the changes, here reported, are in the text in red color (see page 1, lines 42-45; page 2, lines 48-50; lines 68-69, lines 86-99).

Reviewer 2 Report

Nunziato et al. report that a missense variant in NAT9 gene from a family with psoriasis by a multigene panel analysis. This is an interesting paper with the data analyzed in detail. However, there are several concerns which have to be addressed by the authors.

1) Why did you show 6 results (6 waves) of Sanger sequencing in Figure 1-A and 1-B, respectively? I think you should show one wave for each case.

2) If you describe “p.D10N” in Figure 1-A, you should do “p.S43L” but not “p.Ser43Leu” in Figure 1-B.

3) You should describe the information about the father.

Author Response

Point-by-Point

A familial novel putative-pathogenically mutation identified in plaque-psoriasis by a multigene panel analysis

Marcella Nunziato, Anna Balato, Anna Ruocco, Valeria D'Argenio, Roberta Di Caprio, Nicola Balato, Fabio Ayala, Francesco Salvatore

Reviewer #2

Nunziato et al. report that a missense variant in NAT9 gene from a family with psoriasis by a multigene panel analysis. This is an interesting paper with the data analyzed in detail. However, there are several concerns which have to be addressed by the authors.

1) Why did you show 6 results (6 waves) of Sanger sequencing in Figure 1-A and 1-B, respectively? I think you should show one wave for each case.

Reply: We chose to perform direct sequencing (Sanger Sequencing) in our 3 subjects in order to obtain more robust data regarding the substitutions found. The first two electropherograms (EPGs) concern the two strands (FW and RW) of patientPS_4 who was affected by psoriasis. The third and fourth EPGs are from PS_5 and the last two EPGs are from the control healthy subject. This is now better clarified in the figure legend, i.e., as follows: " The electropherograms obtained by Sanger sequencing; for each patient is represented the forward (FW) and the reverse (RW) strands of the same sequence"(see page 4, Figure 1 legend at lines 146-147).

2) If you describe “p.D10N” in Figure 1-A, you should do “p.S43L” but not “p.Ser43Leu” in Figure 1-B.

Thank you for the comment.  We have corrected this in the new figure (Figure 1) in the text (the figure has been replaced).

3) You should describe the information about the father.

Reply: Unfortunately, the father was not available and did not give his informed consent for blood withdrawal for the molecular investigation.

Final Note: As suggested the paper has been accurately revised by a Native-English-Language former Editor of scientific Papers.

Reviewer 3 Report

The presented case provides an interesting about novel putative-pathogenically mutation identified in 2 plaque-psoriasis. Hereby, some major comments to consider the manuscript for publication:

1. Introduction should be extended to include detailed information about psoriasis.

2. Research question should be included at the end of the introduction section

3. Please do consider including a table or figure summarizing the experimental groups, the experimental protocol, dose of infection and references in your work for easier readership.

4. What about molecular identification of the isolates/colonies? You did not confirm your finings and your work is just based on morphology of the colonies?

5. Discussion using your findings and comparing it’s with what was reported in the previous work and the suggested mechanisms underlying this novel putative-pathogenically mutation.

6. Conclusion should be elaborated.

Given the above comments, my suggestion is major revision and manuscript can be considered after addressing my comments.

Author Response

Point-by-Point

A familial novel putative-pathogenically mutation identified in plaque-psoriasis by a multigene panel analysis

Marcella Nunziato, Anna Balato, Anna Ruocco, Valeria D'Argenio, Roberta Di Caprio, Nicola Balato, Fabio Ayala, Francesco Salvatore

Reviewer #3

The presented case provides an interesting about novel putative-pathogenically mutation identified in 2 plaque-psoriasis. Hereby, some major comments to consider the manuscript for publication:

  1. Introduction should be extended to include detailed information about psoriasis.

Reply: Thank you very much for your comments that have enabled us to improve our work. We have largely rewritten the Introduction with more informative sentences and changing some literature references. Furthermore, we have also added a final sentence at the end of the introduction. All the changes, here reported, are in the text in red color (see page 1, lines 42-45; page 2, lines 48-50; lines 68-69, lines 86-99).

  1. Research question should be included at the end of the introduction section

Reply: Thank you we have now included the research question at the end of the Introduction section, i.e., "Herein, we used a 96-multigene panel related to psoriasis risk onset to analyze a family constituted by a mother affected by psoriasis and two daughters, one of whom was also affected by psoriasis while the other healthy daughter served as a family control. The analysis highlighted interesting variants that may be implicated in psoriasis onset, including one in the NAT9 gene. The latter variant is not reported in the ClinVar database nor is it associated to psoriasis in the GWAS catalog and lastly, to our knowledge, it does not appear in any other predisposition studies. Consequently, it appears to be a novel variant related to psoriasis. The aim of this study was to identify new variants that may play a role in the onset of this complex disease." (See page 2, lines 91-99).

  1. Please do consider including a table or figure summarizing the experimental groups, the experimental protocol, dose of infection and references in your work for easier readership.
  2. What about molecular identification of the isolates/colonies? You did not confirm your finings and your work is just based on morphology of the colonies?

Reply: We have added a figure that summarizes the four steps performed in the study to be printed in the Supplementary Material (new figure S1, see page 3 of the Supplemental Material).

Furthermore, we have not used colonies or their morphology herein.

  1. Discussion using your findings and comparing it’s with what was reported in the previous work and the suggested mechanisms underlying this novel putative-pathogenically mutation.

Reply: Unfortunately, knowledge about the variant in the NAT9 gene is very limited.: It is present in dbSNP but not in the ClinVar database. It has never previously been associated with psoriasis (the variant is not present in the GWAS catalog). Unlike the gene which, in some earlier works, it is relevant in MHC antigen presentation, T cell development and it affects the development of autoimmune diseases such as psoriasis due to its role in glycosylation. We have added a new reference, number 31, in which the role of the NAT9 gene for the onset of autoimmune pathologies is mentioned.

  1. Conclusion should be elaborated.

Reply: Thanks, to address this point we have added the following: "In this context, the present study helps to shed light on the possibility that genetic alterations may be the con-cause of the onset of psoriasis or may cooperate for the pathogenesis of the disease itself, together with other multi-factorial factors that characterize this complex disease." (see page 7, lines 261-264).

Final Note: As suggested the paper has been accurately revised by a Native-English-Language former Editor of scientific Papers.

Given the above comments, my suggestion is major revision and manuscript can be considered after addressing my comments.

Round 2

Reviewer 3 Report

The revised version of the manuscript has been improved in light of reviewer's comments. My suggestion is to accept the manuscript in its present form.